# Paths to Equilibrium in Games

**Bora Yongacoglu**
University of Toronto
bora.yongacoglu@utoronto.ca

**Gürdal Arslan**
University of Hawaii at Manoa
gurdal@hawaii.edu

**Lacra Pavel**
University of Toronto
pavel@control.toronto.edu

**Serdar Yüksel**
Queen's University
yuksel@queensu.ca

## Abstract

In multi-agent reinforcement learning (MARL) and game theory, agents repeatedly interact and revise their strategies as new data arrives, producing a sequence of strategy profiles. This paper studies sequences of strategies satisfying a pairwise constraint inspired by policy updating in reinforcement learning, where an agent who is best responding in one period does not switch its strategy in the next period. This constraint merely requires that optimizing agents do not switch strategies, but does not constrain the non-optimizing agents in any way, and thus allows for exploration. Sequences with this property are called satisficing paths, and arise naturally in many MARL algorithms. A fundamental question about strategic dynamics is such: for a given game and initial strategy profile, is it always possible to construct a satisficing path that terminates at an equilibrium? The resolution of this question has implications about the capabilities or limitations of a class of MARL algorithms. We answer this question in the affirmative for normal-form games. Our analysis reveals a counterintuitive insight that reward deteriorating strategic updates are key to driving play to equilibrium along a satisficing path.

## 1 Introduction

Game theory is a mathematical framework for studying strategic interaction between self-interested agents, called players. In an $n$-player normal-form game, each player $i = 1, \cdots, n$, selects a strategy $x^i \in \mathcal{X}^i$ and receives a reward $R^i(x^1, \ldots, x^n)$, which depends on the collective *strategy profile* $\mathbf{x} = (x^1, \ldots, x^n) =: (x^i, \mathbf{x}^{-i})$. Player $i$'s optimization problem is to *best respond* to the strategy $\mathbf{x}^{-i}$ of its counterparts, choosing $x^i \in \mathcal{X}^i$ to maximize $R^i(x^i, \mathbf{x}^{-i})$. Game theoretic models are pervasive in machine learning, appearing in fields such as multi-agent systems [21], multi-objective reinforcement learning [24], and adversarial model training [7], among many others.

In multi-agent reinforcement learning (MARL), players use learning algorithms to revise their strategies in response to the observed history of play, producing a sequence $\{\widehat{\mathbf{x}}_t\}_{t \geq 1}$ in the set of strategy profiles $\mathbf{X} := \mathcal{X}^1 \times \cdots \times \mathcal{X}^n$. Due to the coupled reward structure of multi-agent systems, each player's learning problem involves a moving target: since an individual's reward function depends on the strategies of the others, strategy revision by one agent prompts other agents to revise their own strategies. Convergence analysis of MARL algorithms can therefore be difficult, and the development of tools for such analysis is an important aspect of multi-agent learning theory.

A strategy profile $(x_*^i)_{i=1}^n$ is called a *Nash equilibrium* if all players simultaneously best respond to one another. Nash equilibrium is a concept of central importance in game theory, and the tasks of computing, approximating, and learning Nash equilibrium have attracted enduring attention in theoretical machine learning [47, 27, 14, 42, 17, 26, 31]. Convergence to equilibrium strategies has long been a predominant, but not unique, design goal in MARL [53]. In this paper, we study

38th Conference on Neural Information Processing Systems (NeurIPS 2024).

mathematical structure of normal-form games with the twin objectives of *(i)* better understanding the capabilities or limitations of existing MARL algorithms and *(ii)* producing insights for the design of new MARL algorithms.

A number of MARL algorithms approximate dynamical systems $\{\mathbf{x}_t\}_{t\geq 1}$ on the set of strategy profiles $\mathbf{X}$ in which the next strategy for player $i$ is selected as $x_{t+1}^i = f^i(\mathbf{x}_t)$, where $\mathbf{x}_t = (x_t^1, \ldots, x_t^n)$ is the strategy profile in period $t$. A sampling of such algorithms will be offered shortly. This approach facilitates analysis of the algorithm, as one separately considers the convergence of $\{\mathbf{x}_t\}_{t\geq 1}$ induced by the update functions $\{f^i\}_{i=1}^n$, on one hand, and the approximation of $\{\mathbf{x}_t\}_{t\geq 1}$ by the algorithm's iterates $\{\widehat{\mathbf{x}}_t\}_{t\geq 1}$ on the other. In this work, we consider update functions that satisfy a quasi-rationality condition called *satisficing:* when an agent is best responding, the update rule instructs the agent to continue using this strategy. That is, if $x^i$ is a best response to $\mathbf{x}^{-i}$, then $f^i(x^i, \mathbf{x}^{-i}) = x^i$. This quasi-rationality constraint generalizes the best response update and is desirable for stability of the resulting dynamics, as it guarantees that Nash equilibria are invariant under the dynamics. Moreover, the satisficing condition is only quasi-rational, in that it imposes no constraint on strategy updates when an agent is not best responding, and so allows for exploratory strategy updates. Update rules that incorporate exploratory random search when a strategy is deemed unsatisfactory are common in MARL theory [6, 32, 11, 34].

Our goal is to better understand the capabilities/limitations of MARL algorithms that use the satisficing principle to select successive strategies, potentially augmented with random exploration when an agent is not best responding. Examples include [19, 20, 33, 12, 10, 1] and [52]. Instead of studying a particular collection of strategy update functions, we abstract the problem to the level of sequences in $\mathbf{X}$, which allows us to implicitly account for experimental strategy updates. A sequence $(\mathbf{x}_t)_{t\geq 1}$ of strategy profiles is called a *satisficing path* if, for each player $i$ and time $t$, one has that $x_{t+1}^i = x_t^i$ whenever $x_t^i$ is a best response to $\mathbf{x}_t^{-i}$. The central research question of this paper is such:

> *For a normal-form game $\Gamma$ and an initial strategy profile $\mathbf{x}_1$, is it always possible to construct a satisficing path from $\mathbf{x}_1$ to a Nash equilibrium of the game $\Gamma$?*

Since many MARL algorithms operate using the satisficing principle (or otherwise approximate processes that involve satisficing update rules, e.g. [48]), the resolution of this question has implications for the effectiveness of such MARL algorithms. Indeed, the question has been answered in the affirmative for two-player normal-form games by [19] and for $n$-player symmetric Markov games by [52], and in both classes of games this has directly lead to MARL algorithms with convergence guarantees for approximating equilibria. In addition to removing a theoretical obstacle, positive resolution of this question would establish that *uncoordinated, distributed* random search can effectively assist Nash-seeking algorithms to achieve last-iterate convergence guarantees in a more general class of games than previously possible.

**Contributions.** We give a positive answer to the question above: for any finite $n$-player game $\Gamma$ and any initial strategy profile $\mathbf{x}_1$, there exists a satisficing path beginning at $\mathbf{x}_1$ and ending at a Nash equilibrium of $\Gamma$. This partially answers an open question posed by [52]. We prove this result by analytically constructing a satisficing path from an arbitrary initial strategy profile to a Nash equilibrium. Our approach is somewhat counterintuitive, in that it does not attempt to seek Nash equilibrium by improving the performance of unsatisfied players (players who are not best responding at a given strategy profile), but by updating strategies in a way that *increases* the number of unsatisfied players at each round. This tactic leverages the freedom afforded to unsatisfied players to explore their strategy space and avoids the challenge of cyclical strategy revision that occurs when agents attempt to best respond to their counterparts [37]. This insight provides a new approach to MARL algorithm design beyond the well-structured settings considered in prior work.

**Notation.** We let $\Delta_A$ denote the set of probability measures over a set $A$. For $n \in \mathbb{N}$, we let $[n] := \{1, 2, \ldots, n\}$. For a point $x$, the Dirac measure centered at $x$ is denoted $\delta_x$. When discussing a fixed agent $i$, the remaining collection of agents are called $i$'s counterparts or counterplayers.

**Related Work.** A vast number of MARL algorithms have been proposed for iterative strategy adjustment while playing a game. The most widely studied class of algorithms of this type involve each player running a no-regret algorithm on its own stream of rewards. The celebrated fictitious play algorithm [9] and its descendants are special cases of this class. Although the convergence behavior of fictitious play and its variants has been studied extensively, convergence results are typically available

only for games exhibiting special structural properties amenable to analysis [25, 29, 4, 45, 46]. Indeed, the convergence properties of fictitious play are intimately connected to those of *best response dynamics*, a full information dynamical system evolving in continuous time where the evolution rule for player $i$'s strategy is governed by its best response multi-function. By harnessing such connections, convergence results for fictitious play and a number of other MARL algorithms have been obtained by analyzing the dynamical systems induced by specific update rules [5, 28, 49].

A related line of research considers strategic dynamics defined by strategy update functions, taking the form $x_{t+1}^i = f^i(\mathbf{x}_t)$ in discrete time or an analogous form in continuous time. In the case of deterministic strategy updates, [22] studied strategic dynamics in continuous time and showed that if the strategy update functions, analogous to $f^i$ above, satisfy regularity conditions as well as a desirable property called uncoupledness, by which $f^i$ cannot depend on the reward functions of $i$'s counterplayers, then the resulting dynamics are not Nash convergent in general. These results were recently generalized by [38]. Additional possibility and impossibility results were presented by [2], who studied strategic dynamics in a different setting, where players do not observe counterplayer strategies. Under stochastic strategic dynamics, a number of positive results were obtained by incorporating exogenous randomness into one's strategy update, along with finite recall of recent play [23, 19, 20]. In the regret testing algorithm of [19], players revise their strategies according to whether or not their most recent strategy met a satisfaction criterion: if $x_t^i$ performed within $\epsilon$ of the optimal performance against $\mathbf{x}_t^{-i}$, player $i$ continues using it and picks $x_{t+1}^i = x_t^i$. Otherwise, player $i$ experiments and selects $x_{t+1}^i$ according to a probability distribution over $\mathcal{X}^i$. Conditional strategy updates similar to this have appeared in several other works, such as [12, 10, 11], and the regret testing algorithm has been extended in several ways [20, 1].

A game is said to have the *satisficing paths property* if every initial strategy profile is connected to some equilibrium by a satisficing path. As we discuss in the next section, satisficing paths can be interpreted as a natural generalization of best response paths. Consequently, the problem of identifying games that have the satisficing paths property is a theoretically relevant question analogous to characterizing potential games [41] or determining when a game has the fictitious play property [39, 40]. The concept of satisficing paths was first formalized in [52] in the context of multi-state Markov games, where it was shown that $n$-player symmetric Markov games have the satisficing paths property and this fact could be used to produce a convergent MARL algorithm. However, the core idea of satisficing paths appeared earlier, before this formalization: in the convergence analysis of the regret testing algorithm in [19], it was shown that two-player normal-form games have the satisficing paths property, though this terminology was not used. These earlier works made no claims about the existence of paths in general-sum $n$-player games, which is the focus of this paper.

## 2 Normal-form games

A finite, $n$-player normal-form game $\Gamma$ is described by a list $\Gamma = (n, \mathbf{A}, \mathbf{r})$, where $n$ is the number of players, $\mathbf{A} = \mathbb{A}^1 \times \cdots \times \mathbb{A}^n$ is a finite set of action profiles, and $\mathbf{r} = (r^i)_{i \in [n]}$ is a collection of reward functions, where $r^i : \mathbf{A} \to \mathbb{R}$ describes the reward of player $i$ as a function of the action profile. The $i^{\text{th}}$ component of $\mathbf{A}$ is player $i$'s action set $\mathbb{A}^i$.

**Description of play.** Each player $i \in [n]$ selects a probability vector $x^i \in \Delta_{\mathbb{A}^i}$ and then selects its action $a^i$ according to $a^i \sim x^i$. The vector $x^i$ is called player $i$'s mixed strategy, and we denote player $i$'s set of mixed strategies by $\mathcal{X}^i := \Delta_{\mathbb{A}^i}$. Players are assumed to select their actions without observing one another's actions, and the collection of actions $\{a^i : i \in [n]\}$ is assumed to be mutually independent. The set of mixed strategy profiles is denoted $\mathbf{X} := \mathcal{X}^1 \times \cdots \mathcal{X}^n$. After the action profile $\mathbf{a} = (a^1, \ldots, a^n)$ is selected, each player $i$ receives reward $r^i(\mathbf{a})$.

Player $i$'s performance criterion is its expected reward, defined for each strategy profile $\mathbf{x} \in \mathbf{X}$ as

$$R^i(x^i, \mathbf{x}^{-i}) = \mathbb{E}_{\mathbf{a} \sim \mathbf{x}} \left[ r^i(a^1, \ldots, a^n) \right],$$

where $\mathbb{E}_{\mathbf{a} \sim \mathbf{x}}$ signifies that $a^j \sim x^j$ for each player $j \in [n]$ and we have used the convention that $\mathbf{x} = (x^i, \mathbf{x}^{-i})$ and $\mathbf{x}^{-i} = (x^1, \ldots, x^{i-1}, x^{i+1}, \ldots, x_n)$. Since player $i$'s objective depends on the strategies of its counterplayers, the relevant optimality notion is that of ($\epsilon$-) best responding.

**Definition 1.** *A mixed strategy $x_*^i \in \mathcal{X}^i$ is called an $\epsilon$-best response to the strategy $\mathbf{x}^{-i} \in \mathbf{X}^{-i}$ if*

$$R^i(x_*^i, \mathbf{x}^{-i}) \geq R^i(x^i, \mathbf{x}^{-i}) - \epsilon \quad \forall x^i \in \mathcal{X}^i.$$

The standard solution concept for $n$-player normal form games is that of ($\epsilon$-) Nash equilibrium, which entails a situation in which all players are simultaneously ($\epsilon$-) best responding to one another.

**Definition 2.** *For $\epsilon \geq 0$, a strategy profile $\mathbf{x}_* = (x_*^i, \mathbf{x}_*^i) \in \mathbf{X}$ is called an $\epsilon$-Nash equilibrium if, for every player $i \in [n]$, $x_*^i$ is an $\epsilon$-best response to $\mathbf{x}_*^{-i}$.*

Putting $\epsilon = 0$ above, one recovers the classical definitions of *best responding* and *Nash equilibrium*. For any $\epsilon \geq 0$, the set of $\epsilon$-best responses to a strategy $\mathbf{x}^{-i}$ is denoted $\mathrm{BR}_\epsilon^i(\mathbf{x}^{-i}) \subseteq \mathcal{X}^i$.

## 2.1 Satisficing Paths

We now present the concept of satisficing paths as generalized best response paths.

**Definition 3.** *A sequence of strategy profiles $(\mathbf{x}_t)_{t \geq 1}$ in $\mathbf{X}$ is called a best response path if, for every $t \geq 1$ and every player $i \in [n]$, we have*

$$x_{t+1}^i = \begin{cases} x_t^i, & \text{if } x_t^i \in \mathrm{BR}_0^i(\mathbf{x}_t^{-i}), \\ \text{some } x_\star^i \in \mathrm{BR}_0^i(\mathbf{x}_t^{-i}), & \text{else.} \end{cases}$$

The preceding definition of best response paths can be relaxed in several ways, and such relaxations are often desirable to avoid non-convergent cycling behavior (see [37] for an example). A common relaxation involves synchronizing players or incorporating inertia, so that only a subset of players switch their strategies at a given time, which can be help achieve coordination in cooperative settings [32, 48, 51]. Beyond cooperative settings, the use of best response dynamics to seek Nash equilibrium may not be justified. In purely adversarial settings, for instance, best response paths cycle and fail to converge [3], and some alternative strategic dynamics are needed to drive play to equilibrium. Consider the following generalization of the best response update:

$$x_{t+1}^i = \begin{cases} x_t^i, & \text{if } x_t^i \in \mathrm{BR}_0^i(\mathbf{x}_t^{-i}), \\ f^i(x_t^i, \mathbf{x}_t^{-i}) & \text{else.} \end{cases}$$

The update defined above is characterized by a "win–stay, lose–shift" principle [11, 44], which only constrains the player to continue using a strategy when it is optimal. On the other hand, the player is not forced to use a best response when $x_t^i \notin \mathrm{BR}_0^i(\mathbf{x}_t^{-i})$, and may experiment with suboptimal responses according to a function $f^i : \mathbf{X} \to \mathcal{X}^i$.[1] Allowing the function $f^i$ to be any function from $\mathbf{X}$ to $\mathcal{X}^i$, one generalizes best response updates and obtains a much larger set of sequences $(\mathbf{x}_t)_{t \geq 1}$ and greater flexibility to approach equilibrium from new directions. This motivates the following definition of satisficing paths.

**Definition 4.** *A sequence of strategy profiles $(\mathbf{x}_t)_{t=1}^T$, where $T \in \mathbb{N} \cup \{\infty\}$, is called a satisficing path if it satisfies the following pairwise satisfaction constraint for any player $i \in [n]$ and any $t$:*

$$x_t^i \in \mathrm{BR}_0^i(\mathbf{x}_t^{-i}) \Rightarrow x_{t+1}^i = x_t^i. \tag{1}$$

The intuition behind satisficing paths is that they are the result of an iterative search process in which players settle upon finding an optimal strategy (i.e. a best response to the strategies of counterplayers) but are free to explore different strategies when they are not already behaving optimally. Note, however, that the definition above is merely a formal property of sequences of strategy profiles in $\mathbf{X}$ and is agnostic to how a satisficing path is produced. The latter point will be important in the coming sections, where we analytically obtain a particular satisficing path as part of an existence proof.

We note that Condition (1) constrains only optimizing players. It does not mandate a particular update for the so-called unsatisfied player $i$, for whom $x_t^i \notin \mathrm{BR}_0^i(\mathbf{x}_t^{-i})$. In particular, $x_{t+1}^i$ can be any strategy without restriction, and $x_{t+1}^i \notin \mathrm{BR}_0^i(\mathbf{x}_t^{-i})$ is allowed. In addition to best response paths, constant sequences $(\mathbf{x}_t)_{t \geq 1}$ with $\mathbf{x}_t \equiv \mathbf{x}$ are always satisficing paths, even when $\mathbf{x}$ is not a Nash equilibrium. Moreover, since arbitrary strategy revisions are allowed when a player is unsatisfied, if $\mathbf{x}_1 \in \mathbf{X}$ is a strategy profile for which all players are unsatisfied, then $(\mathbf{x}_1, \mathbf{x}_2)$ is a satisficing path for any $\mathbf{x}_2 \in \mathbf{X}$.

---

[1] As a special case, $f^i$ may simply be a best response selector, recovering the best response update.

**Definition 5.** *The game $\Gamma$ has the* satisficing paths property *if for any $\mathbf{x}_1 \in \mathbf{X}$, there exists a satisficing path $(\mathbf{x}_1, \mathbf{x}_2, \dots)$ such that, for some finite $T = T(\mathbf{x}_1)$, the strategy profile $\mathbf{x}_T$ is a Nash equilibrium.*[2]

Satisficing paths were initially formalized in [52], where it was proved that two-player games and $n$-player symmetric games have the satisficing paths property. However, whether general-sum $n$-player games have the satisficing paths property was left as an open question. We answer this open question in Theorem 1, presented in the next section.

## 3 Existence of paths in normal-form games

**Theorem 1.** *Any finite normal-form game $\Gamma$ has the satisficing paths property.*

**Proof sketch.** Before presenting the formal proof, we describe the intuition of its main argument. In the proof of Theorem 1, we construct a satisficing path from an arbitrary initial strategy $\mathbf{x}_1$ to a Nash equilibrium by repeatedly switching the strategies of unsatisfied players in a way that grows the set of *unsatisfied* players after the update. Once the set of unsatisfied players is maximal, we argue that a Nash equilibrium can be reached in one step by switching the strategies of the unsatisfied players. The final point represents the main technical challenge in the proof, as switching the strategies of unsatisfied players changes the objective functions for the previously satisfied players. We address this challenge by showing the existence of a Nash equilibrium on the boundary of a strategy subset in which previously satisfied players remain satisfied.

To give the complete proof, we will require some additional notation, detailed below, and some supporting results, detailed in Appendix A and Appendix B.

**Additional notation.** We require notation for the following sets, defined for any $\mathbf{x} \in \mathbf{X}$:

$$\mathrm{Sat}(\mathbf{x}) := \left\{i \in [n] : x^i \in \mathrm{BR}_0^i(\mathbf{x}^{-i})\right\}, \quad \text{and} \quad \mathrm{UnSat}(\mathbf{x}) := [n] \setminus \mathrm{Sat}(\mathbf{x}).$$

A player in $\mathrm{Sat}(\mathbf{x}) \subseteq [n]$ is called *satisfied* (at $\mathbf{x}$), and a player in $\mathrm{UnSat}(\mathbf{x})$ is called *unsatisfied* (at $\mathbf{x}$). For $\mathbf{x} \in \mathbf{X}$, we also define

$$\mathrm{Access}(\mathbf{x}) := \left\{\mathbf{y} \in \mathbf{X} : y^i = x^i,\ \forall i \in \mathrm{Sat}(\mathbf{x})\right\}.$$

$\mathrm{Access}(\mathbf{x})$ is the subset of strategies that are accessible from strategy $\mathbf{x}$, to mean one can obtain strategy $\mathbf{y} \in \mathrm{Access}(\mathbf{x}) \subseteq \mathbf{X}$ from $\mathbf{x}$ by switching (at most) the strategies of players who were unsatisfied at $\mathbf{x}$. We define a subset $\mathrm{NoBetter}(\mathbf{x}) \subseteq \mathrm{Access}(\mathbf{x})$ as

$$\mathrm{NoBetter}(\mathbf{x}) := \{\mathbf{y} \in \mathrm{Access}(\mathbf{x}) : \mathrm{UnSat}(\mathbf{x}) \subseteq \mathrm{UnSat}(\mathbf{y})\}$$
$$= \{\mathbf{y} \in \mathrm{Access}(\mathbf{x}) | \forall i \in \mathrm{UnSat}(\mathbf{x}),\ i \in \mathrm{UnSat}(\mathbf{y})\},$$

The set $\mathrm{NoBetter}(\mathbf{x})$ consists of strategies $\mathbf{y}$ that are accessible from $\mathbf{x}$ and also fail to improve the status of players who were previously unsatisfied. The set name $\mathrm{NoBetter}(\mathbf{x})$ is chosen to suggest that the players unsatisfied at $\mathbf{x}$ are not better off at $\mathbf{y} \in \mathrm{NoBetter}(\mathbf{x})$, since they are unsatisfied at both $\mathbf{x}$ and $\mathbf{y}$. We observe $\mathbf{x} \in \mathrm{NoBetter}(\mathbf{x})$, hence $\mathrm{NoBetter}(\mathbf{x})$ is non-empty.

Finally, we define a set $\mathrm{Worse}(\mathbf{x}) \subseteq \mathrm{NoBetter}(\mathbf{x})$ as

$$\mathrm{Worse}(\mathbf{x}) := \{\mathbf{y} \in \mathrm{NoBetter}(\mathbf{x}) : \mathrm{UnSat}(\mathbf{x}) \subsetneq \mathrm{UnSat}(\mathbf{y})\}$$
$$= \{\mathbf{y} \in \mathrm{NoBetter}(\mathbf{x}) | \exists i \in \mathrm{Sat}(\mathbf{x}) : i \in \mathrm{UnSat}(\mathbf{y})\}.$$

The set $\mathrm{Worse}(\mathbf{x})$ consists of strategies that are accessible from $\mathbf{x}$, that leave all previously unsatisfied players unsatisfied, and flip at least one previously satisfied player to being unsatisfied. In particular, if $\mathbf{y} \in \mathrm{Worse}(\mathbf{x})$, this means $|\mathrm{UnSat}(\mathbf{y})| \geq |\mathrm{UnSat}(\mathbf{x})| + 1$. We observe that $\mathrm{Worse}(\mathbf{x})$ may be empty, and $\mathrm{Worse}(\mathbf{x}) \subseteq \mathrm{NoBetter}(\mathbf{x}) \subseteq \mathrm{Access}(\mathbf{x})$.

---

[2]A more general definition, involving $\epsilon \geq 0$ best responding and strategy subsets was studied in [52]. In this paper, we consider true optimality and no strategic constraints, which additionally aids clarity.

### 3.1 Proof of Theorem 1

**Remark 1.** *In the proof below, we analytically construct a path from $\mathbf{x}_1$ to a Nash equilibrium. The process of selecting strategies $\mathbf{x}_1, \mathbf{x}_2, \cdots$ and switching the component strategy of each player is done centrally, by the analyst, and should not be interpreted as a learning algorithm.*

*Proof.* Let $\mathbf{x}_1 \in \mathbf{X}$ be any initial strategy profile. We must produce a satisficing path of finite length terminating at a Nash equilibrium. Equivalently, we must produce a sequence $\mathbf{x}_1, \ldots, \mathbf{x}_T$ with $\mathbf{x}_{t+1} \in \text{Access}(\mathbf{x}_t)$ for each $t$ and $\mathbf{x}_T$ a Nash equilibrium, where the length $T$ may depend on $\mathbf{x}_1$. In the trivial case that $\mathbf{x}_1$ is a Nash equilibrium, we put $T = 1$. The remainder of this proof focuses on the non-trivial case, where $\mathbf{x}_1$ is not a Nash equilibrium.

To begin, we produce a satisficing path $\mathbf{x}_1, \ldots, \mathbf{x}_k$ as follows. We put $t = 1$, and while both $\text{Sat}(\mathbf{x}_t) \neq \varnothing$ and $\text{Worse}(\mathbf{x}_t) \neq \varnothing$, we arbitrarily fix $\mathbf{x}_{t+1} \in \text{Worse}(\mathbf{x}_t)$ and increment $t \leftarrow t + 1$. By construction, we have

$$\varnothing \neq \text{UnSat}(\mathbf{x}_1) \subsetneq \cdots \subsetneq \text{UnSat}(\mathbf{x}_t) \subsetneq \text{UnSat}(\mathbf{x}_{t+1})$$

for each non-terminal iteration $t$, where the inequality holds because $\mathbf{x}_1$ is not a Nash equilibrium. Thus, the number of unsatisfied players is strictly increasing along this satisficing path. Since the number of unsatisfied players is bounded above by $n$, and since we have assumed $|\text{UnSat}(\mathbf{x}_1)| \geq 1$, this process terminates in at most $n - 1$ steps. Letting $k$ denote the terminal index of this process, we have $k \leq n - 1$.

By the construction of the path $(\mathbf{x}_1, \ldots, \mathbf{x}_k)$, (at least) one of the following holds at index $k$: either $\text{Sat}(\mathbf{x}_k) = \varnothing$ or $\text{Worse}(\mathbf{x}_k) = \varnothing$. In other words, either no player is satisfied at $\mathbf{x}_k$, or there is no accessible strategy that grows the subset of unsatisfied players.

**Case 1:** $\text{Sat}(\mathbf{x}_k) = \varnothing$, and all players are unsatisfied at $\mathbf{x}_k$. In this case, we may switch the strategy of each player $i \in [n]$ to any successor strategy. That is, $\text{Access}(\mathbf{x}_k) = \mathbf{X}$. We fix an arbitrary Nash equilibrium $\mathbf{z}_\star$, put $\mathbf{x}_{k+1} = \mathbf{z}_\star$, and let $T = k + 1$. Then, $(\mathbf{x}_1, \ldots, \mathbf{x}_T)$ is a satisficing path terminating at equilibrium.

**Case 2:** $\text{Sat}(\mathbf{x}_k) \neq \varnothing$ and $\text{Worse}(\mathbf{x}_k) = \varnothing$. In this case, there are no accessible strategies that strictly grow the set of unsatisfied players.

Since $\text{Worse}(\mathbf{x}_k) = \varnothing$, the following holds: for any strategy $\mathbf{y} \in \text{NoBetter}(\mathbf{x}_k)$ and any satisfied player $i \in \text{Sat}(\mathbf{x}_k)$, we have that $i \in \text{Sat}(\mathbf{y})$. (Otherwise, if $i \in \text{UnSat}(\mathbf{y})$, then $\mathbf{y} \in \text{Worse}(\mathbf{x}_k)$, since it flipped a satisfied player. But this contradicts the emptiness of $\text{Worse}(\mathbf{x}_k)$.)

We now argue that there exists a strategy profile $\mathbf{x}_\star$ accessible from $\mathbf{x}_k$ such that all players unsatisfied at $\mathbf{x}_k$ are satisfied at $\mathbf{x}_\star$. That is, there exists an accessible strategy $\mathbf{x}_\star \in \text{Access}(\mathbf{x}_k)$ such that

$$\text{UnSat}(\mathbf{x}_k) \subset \text{Sat}(\mathbf{x}_\star). \tag{2}$$

To see that such a strategy $\mathbf{x}_\star$ exists, note that fixing the strategies of the $m$ players satisfied at $\mathbf{x}_k$ defines a new game, say $\tilde{\Gamma}$, with $n - m$ players, and the new game $\tilde{\Gamma}$ admits a Nash equilibrium $\tilde{\mathbf{x}}_\star = (\tilde{x}_\star^i)_{i \in \text{UnSat}(\mathbf{x}_k)}$. We extend $\tilde{\mathbf{x}}_\star$ to be a strategy profile in the larger game $\Gamma$ by putting $x_\star^i = x_k^i$ for players $i \in \text{Sat}(\mathbf{x}_k)$ while putting $x_\star^j = \tilde{x}_\star^j$ for players $j \in \text{UnSat}(\mathbf{x}_k)$. By construction, we have that $x_\star^j \in \text{BR}_0^j(\mathbf{x}_\star^{-j})$ for each $j \in \text{UnSat}(\mathbf{x}_k)$, so (2) holds.

From (2), it is clear that $\mathbf{x}_\star \notin \text{NoBetter}(\mathbf{x}_k)$, since $\text{NoBetter}(\mathbf{x}_k)$ consists of strategies accessible from $\mathbf{x}_k$ in which unsatisfied agents remain unsatisfied, while the previously unsatisfied agents are satisfied at $\mathbf{x}_\star$. We now state a key technical lemma, which asserts that although $\mathbf{x}_\star$ does not belong to $\text{NoBetter}(\mathbf{x}_k)$, it is a limit point of this set. A proof of Lemma 1 given in Appendix B.

**Lemma 1.** *If $\text{Worse}(\mathbf{x}_k) = \varnothing$, then there exists a sequence $\{\mathbf{y}_t\}_{t=1}^\infty$, with $\mathbf{y}_t \in \text{NoBetter}(\mathbf{x}_k)$ for each $t$, such that $\lim_{t \to \infty} \mathbf{y}_t = \mathbf{x}_\star$.*

We will argue that $\mathbf{x}_\star$ is a Nash equilibrium for the original game $\Gamma$. For each player $i \in [n]$, we introduce a function $F^i : \mathbf{X} \to \mathbb{R}$ given by $F^i(x^i, \mathbf{x}^{-i}) = \max_{a^i \in \mathbb{A}^i} R^i(\delta_{a^i}, \mathbf{x}^{-i}) - R^i(x^i, \mathbf{x}^{-i})$, for each $\mathbf{x} = (x^i, \mathbf{x}^{-i}) \in \mathbf{X}$. The functions $\{F^i\}_{i=1}^n$ have the following useful properties, which are well known [35], and are summarized in Appendix A. For each player $i \in [n]$: (a) $F^i$ is continuous

on $\mathbf{X}$; (b) $F^i(\mathbf{x}) \geq 0$ for all $\mathbf{x} \in \mathbf{X}$; (c) for any $\mathbf{x}^{-i} \in \mathbf{X}^{-i}$, a strategy $x^i$ is a best response to $\mathbf{x}^{-i}$ if and only if $F^i(x^i, \mathbf{x}^{-i}) = 0$.

Let $(\mathbf{y}_t)_{t=1}^\infty$ be a sequence in $\mathrm{NoBetter}(\mathbf{x}_k)$ converging to $\mathbf{x}_\star$, which exists by Lemma 1. For any previously satisfied player $i \in \mathrm{Sat}(\mathbf{x}_k)$, since $\mathrm{Worse}(\mathbf{x}_k) = \varnothing$ and $\mathbf{y}_t \in \mathrm{NoBetter}(\mathbf{x}_k)$, from a previous observation, we have that $i \in \mathrm{Sat}(\mathbf{y}_t)$. Equivalently, $x_k^i \in \mathrm{BR}_0^i(\mathbf{y}_t^{-i})$. Re-writing this using the function $F^i$ and the notation $y_t^i = x_k^i$ for satisfied players $i \in \mathrm{Sat}(\mathbf{x}_k)$, we have $F^i(y_t^i, \mathbf{y}_t^{-i}) = 0$ for all $t \in \mathbb{N}$ and for any $i \in \mathrm{Sat}(\mathbf{x}_k)$. By continuity of $F^i$, we have

$$0 = \lim_{t\to\infty} F^i(\mathbf{y}_t) = F^i\left(\lim_{t\to\infty} \mathbf{y}_t\right) = F^i(\mathbf{x}_\star),$$

establishing that player $i$ is satisfied at $\mathbf{x}_\star$, and thus that $\mathrm{Sat}(\mathbf{x}_k) \subset \mathrm{Sat}(\mathbf{x}_\star)$. Then, by (2), we had $\mathrm{UnSat}(\mathbf{x}_k) \subset \mathrm{Sat}(\mathbf{x}_\star)$, hence $\mathrm{Sat}(\mathbf{x}_\star) = [n]$, and $\mathbf{x}_\star$ is a Nash equilibrium accessible from $\mathbf{x}_k$. We put $T = k + 1$ and $\mathbf{x}_T = \mathbf{x}_\star$, which completes the proof, since $(\mathbf{x}_1, \ldots, \mathbf{x}_T)$ is a satisficing path terminating at a Nash equilibrium. $\qquad\square$

## 3.2 Algorithmic insights from the proof of Theorem 1

When coupled with a MARL algorithm that uses an exploratory satisficing strategy update, play will be driven along satisficing paths. Theorem 1 shows that for any starting strategy profile, some such path connects the strategy profile to an equilibrium, and so a sufficiently exploratory strategy update may drive play to equilibrium along a satisficing path. This offers important insights for the design of MARL algorithms. The first takeaway from Theorem 1 is that play can be driven to equilibrium by changing only the strategies of those players who are not best responding. In particular, this means that a satisfied agent does not need to continue updating its strategy after it becomes satisfied. As we will discuss in the next section, this property is helpful in distributed and decentralized multi-agent systems, where agents are able to assess whether they are satisfied but may not be able to assess whether the overall system is at equilibrium.

A second, more subtle takeaway comes from the proof of Theorem 1 and relates to the unorthodox and counterintuitive exploration scheme used to drive play to equilibrium. In the proof, one sees that suboptimal—and perhaps even *reward-deteriorating*—strategic updates were key to driving play to equilibrium along a satisficing path. As we outline below, this construction runs against the conventional approaches to designing MARL algorithms, and it can be used to avoid common pitfalls of MARL algorithms such as cyclical behavior.

At a high level, many existing multi-agent learning algorithms update the strategy parameter in a *reward-improving* direction at each step. A related approach, described earlier, increments the strategy parameter in a regret-minimizing direction, which has a similar effect. While such algorithms are sensible from the point of view of a single self-interested individual, they may fail to drive play to a Nash equilibrium when all players adopt similar algorithms [36, 18, 37]. To address this non-convergence issue, one recurring algorithmic modification involves manipulating step sizes, either with a mixture of fast agents and slow agents [13] or with each individual varying its step sizes according to its performance [8]. However, such approaches only come with provable convergence guarantees in select subclasses of games with exploitable structure. In instances where step size manipulation does not (or cannot) yield convergence, the analysis of Theorem 1 may offer an alternative route to algorithm modification.

With these two takeaways in mind, we envision at least two design principles that will be useful for future MARL algorithms. First, strategic updating may incorporate some measure of randomness when a player is not satisfied. This principle has been previously used with some success, but comes with a drawback relating to complexity. A second principle, which we believe to be new, leverages the second takeaway above, involving counterintuitive path construction: players may alternate between reward-improving periods (during which strategy updates are done in a conventional way that improves the agent's reward) and suboptimal periods (during which reward-deteriorating and/or random strategy updates may be used). The timing of such periods or the extent of the randomness in strategic updates may be made to depend on whether cycles in the strategy iterates were detected. By incorporating suboptimal exploration in an adaptive manner, a MARL algorithm can break cycles as needed but rely on conventional algorithms the remainder of the time.

## 4 Discussion

**Extension to Markov games**

This paper focused on normal-form games with finitely many actions per player due to the central position that normal-form games occupy in game theory. Indeed, insights and intuition developed in normal-form games are helpful for understanding more complex models of strategic interaction. Of special note, finite normal-form games can be generalized to model dynamic strategic environments where rewards and environmental parameters evolve over time according to the history of play. We now describe the extension of Theorem 1 to Markov games, one generalization of finite normal-form games that is a popular model in MARL. Due to space limitations, a formal model for Markov games is postponed to Appendix C.

In an $n$-player Markov game, agents interact across discrete time. Each agent $i \in [n]$ observes a sequence of state variables $\{s_t\}_{t \geq 1}$ taking values in a finite state space $\mathcal{S}$ and selects a sequence of actions $\{a_t^i\}_{t \geq 1}$ taking values in a finite action set $\mathbb{A}^i$. In this dynamic model, player $i$'s reward in period $t$, denoted $r_t^i = r^i(s_t, \mathbf{a}_t)$, depends on both the action profile $\mathbf{a}_t$ and also on the state $s_t$. The state process evolves according to a (jointly controlled) transition probability function $\mathcal{T}$ as $s_{t+1} \sim \mathcal{T}(\cdot | s_t, \mathbf{a}_t)$. Rewards are discounted across time using a discount factor $\gamma \in (0, 1)$, and player $i$ attempts to maximize its expected $\gamma$-discounted return. In this generalization of finite normal-form games, *policies* (defined as mappings from states to probability distributions over actions) generalize mixed strategies, and the solution concept of *Markov perfect equilibrium* refines the concept of Nash equilibrium and serves as a popular stability objective for MARL algorithm designers [53].

Partial results for multi-state Markov games have previously been obtained in special classes of games and used to produce MARL algorithms [52]. The analysis presented in this paper uses a rather different approach that seems promising for extending those results. In the proof of Theorem 1, we used functions $\{F^i\}_{i=1}^n$ to characterize best responding in a finite normal-form game. In fact, analogous functions can also be obtained for policies in multi-state Markov games, and these functions satisfy the same desired properties invoked in the proof of Theorem 1 (c.f. [52, Lemmas 2.10-2.13]). For this reason, and due to the central role of continuity in our proof, it seems likely that Theorem 1 can be extended to general-sum Markov games. However, one aspect of the extension remains open, namely the generalization of Lemma 1. In Appendix C, we describe the issue that precludes direct generalization of our normal-form proof of Lemma 1, but we note that this appears to be related only to the proof technique rather than a fundamental obstacle to the generalization.

**On decentralized learning**

Multi-agent reinforcement learning algorithms based on the "win–stay, lose–shift" principle characteristic of satisficing paths are especially well suited to decentralized applications, since players are often able to estimate the performance of their current strategy as well as the performance of an optimal strategy, even under partial information. In decentralized problems, coordinated search of the set $\mathbf{X}$ of strategy profiles for a Nash equilibrium is typically infeasible, and players must select successor strategies in a way the depends only on quantities that can be locally accessed or estimated.

For instance, consider a trivial coordinated search method, where player $i$ selects $x_{t+1}^i$ uniformly at random from $\mathcal{X}^i$ whenever $\mathbf{x}_t$ was not a Nash equilibrium and selects $x_{t+1}^i = x_t^i$ only when $\mathbf{x}_t$ is a Nash equilibrium. This process is clearly ill suited to decentralized applications, because player $i$'s strategy update depends on both a locally estimable condition (whether player $i$ is best responding to $\mathbf{x}_t^{-i}$) as well as a condition that cannot be locally estimated (whether another player $j \neq i$ is best responding to $\mathbf{x}_t^{-j}$.) The satisfaction (win–stay) constraint plays a key role as a *local* stopping condition for satisficing paths, and rules out coordinated search of the set $\mathbf{X}$ such as the trivial update outlined above. Examples of decentralized or partially decentralized learning algorithms leveraging satisficing paths in their analysis include [19, 33, 1, 52]. The analytic results of this paper suggest that algorithms such as these can be extended to wider classes of games and enjoy equilibrium guarantees under different informational constraints on the players.

**On complexity and dynamics**

In Theorem 1, we showed that for any finite $n$-player normal-form game $\Gamma$ and any initial strategy profile $\mathbf{x}_1 \in \mathbf{X}$, there exists a satisficing path $\mathbf{x}_1, \ldots, \mathbf{x}_T$ of finite length $T = T(\mathbf{x}_1)$ terminating at a Nash equilibrium $\mathbf{x}_T$. From the proof of Theorem 1, one makes the following observations. First, the length of such a path can be uniformly bounded above as $T(\mathbf{x}_1) \leq n$. Second, there exists a collection of strategy update functions $\{f_\Gamma^i : \mathbf{X} \to \mathcal{X}^i \big| i \in [n]\}$ whose joint orbit is the satisficing path described by the proof of Theorem 1. That is, $f_\Gamma^i(\mathbf{x}_t) = x_{t+1}^i$ for each player $i \in [n]$, every $0 \leq t \leq T - 1$, and every $\mathbf{x}_1 \in \mathbf{X}$, where $x_t^i$ is player $i$'s component of $\mathbf{x}_t$ in the satisficing path initialized at $\mathbf{x}_1$.

The proof of Theorem 1 is semi-constructive. At each step along the path, we describe how the next strategy profile should be picked (e.g. $\mathbf{x}_{t+1} \in \text{Worse}(\mathbf{x}_t)$), but we do not suggest an algorithm for computing it. In at least one place, namely Case 1 where we put $\mathbf{x}_T := \mathbf{z}_\star$, the path construction involves moving jointly to a Nash equilibrium in one step. The computational complexity of such a step is prohibitive [15], underscoring that ours is an analytical existence result rather than a computational prescription.

Although we have shown that there exists a discrete-time dynamical system on $\mathbf{X}$ that converges to Nash equilibrium in $n$ steps and can be characterized by update functions $\{f_\Gamma^i\}_{i=1}^n$, we note that our possibility result does not contradict the impossibility results of [22, 2] or [38]. In particular, the functions $\{f_\Gamma^i\}_{i=1}^n$ need not be (and usually will not be) continuous, violating the regularity conditions of [22] and [38], and furthermore the functions $\{f_\Gamma^i\}_{i=1}^n$ depend crucially on the game $\Gamma$ in a way that violates the uncoupledness conditions of [22] and [2].

**Open questions and future directions**

Several interesting questions about satisficing paths remain open. We now briefly describe some that we find especially practical or theoretically relevant.

While this paper dealt with satisficing paths defined using a best responding constraint, the original definition was stated using an $\epsilon$-best responding constraint, according to which a player who was $\epsilon$-best responding was not allowed to switch its strategy. Putting $\epsilon = 0$, one recovers the definition used here, but one may also select $\epsilon > 0$, which can be desirable to accommodate for estimation error in multi-agent reinforcement learning applications. The added constraint reduces freedom to switch strategies, and thus makes it more challenging to construct paths starting from a given strategy profile. On the other hand, the collection of Nash equilibria is a strict subset of the set of $\epsilon$-Nash equilibria, and one can attempt to guide the process to a different terminal point in a larger set. At this time, it is not clear to us whether the main result of this paper holds for small $\epsilon > 0$. It is clear, however, that the proof technique used here will have to be modified, since we have relied on Lemma 1, whose proof involved an indifference condition and invoked the fundamental theorem of algebra, and relaxing to $\epsilon > 0$ would render such an argument ineffective.

A second interesting question for future work is whether multi-state Markov games with $n > 2$ players have the satisficing paths property. The case with $n = 2$ was resolved by [52], but the proof technique used there did not generalize to $n \geq 3$. By contrast, our proof technique readily accommodates any number of players, but is designed for stateless normal-form games. Our proof used multi-linearity of the expected reward functions $\{R^i\}_{i=1}^n$, which does not generally hold in the multi-state setting.

In this work, satisficing paths were defined in a way that allowed an unsatisfied player $i$ to change its strategy to any strategy in its set $\mathcal{X}^i$, without constraint. This is interesting in many problems where the set of strategies can be explicitly and directly parameterized, but may be unrealistic in games where the set of strategies is poorly understood or in which a player can effectively represent only a subset of its strategies $\mathcal{Y}^i \subsetneq \mathcal{X}^i$. In such games, the question more relevant for algorithm design is whether the game admits satisficing paths to equilibrium within the restricted subset $\mathcal{Y}^1 \times \cdots \times \mathcal{Y}^n$. This point was implicitly appreciated by both [19] and [20] and explicitly noted in [52]. Some negative results were recently established in [50] for games admitting pure strategy Nash equilibrium when randomized action selection was not allowed and the constrained set was given by $\mathcal{Y}^i = \mathbb{A}^i$, underscoring the importance of the topology of the sets appearing in the proof of Theorem 1.

## 5  Conclusion

Satisficing paths can be interpreted as a natural generalization of best response paths in which players may experimentally select their next strategy in periods when they fail to best respond to their counterplayers. While (inertial) best response dynamics drive play to equilibrium in certain well-structured classes of games, such as potential games and weakly acyclic games [16], the constraint of best responding limits the efficacy of these dynamics in games with cycles in the best response graph [43]. In such games, best response paths leading to equilibrium do not exist, and multi-agent reinforcement learning algorithms designed to produce such paths will not lead to equilibrium.

In this paper, we have shown that every finite normal-form game enjoys the satisficing paths property. By relaxing the best response constraint for unsatisfied players, one ensures that paths to equilibrium exist from any initial strategy profile. Multi-agent reinforcement learning algorithms designed to produce satisficing paths, rather than best response paths, thus do not face the same fundamental obstacle of algorithms based on best responding. While algorithms based on satisficing have previously been developed for two-player games normal-form games, symmetric Markov games, and several other subclasses of games, the findings of this paper suggest that similar algorithms can be devised for the wider class of $n$-player general-sum normal-form games.

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

## Appendix: Proofs of technical lemmas

We now discuss the properties of the auxiliary functions $\{F^i : i \in [n]\}$ that were used in the proof of Theorem 1, and we prove Lemma 1.

We remark that for each player $i \in [n]$, we identify their set of mixed strategies $\mathcal{X}^i = \Delta_{\mathbb{A}^i}$ with the probability simplex in $\mathbb{R}^{\mathbb{A}^i}$. Thus, $\mathcal{X}^i$ inherits the Euclidean metric from $\mathbb{R}^{|\mathbb{A}^i|}$. Neighbourhoods and limits in $\mathcal{X}^i$ (or its subsets) are defined with respect to this metric. Similarly, we inherit a Euclidean metric for $\mathbf{X}$. For $\zeta > 0$, we let $N_\zeta(\mathbf{x})$ denote the $\zeta$-neighbourhood of the strategy profile $\mathbf{x} \in \mathbf{X}$.

# A    Properties of the auxiliary functions

We begin by discussing the properties of the auxiliary functions $\{F^i : i \in [n]\}$, as they are relevant to characterizing best responses. The facts below are well known. For a reference, see the text of [35].

Recall that for each player $i \in [n]$, the function $F^i : \mathbf{X} \to \mathbb{R}$ is defined as

$$F^i(x^i, \mathbf{x}^{-i}) = \max_{a^i \in \mathbb{A}^i} R^i(\delta_{a^i}, \mathbf{x}^{-i}) - R^i(x^i, \mathbf{x}^{-i}), \quad \forall \mathbf{x} \in \mathbf{X}.$$

We now show that for any $i \in [n]$, the following hold:

  a. $F^i$ is continuous on $\mathbf{X}$,
  b. $F^i(\mathbf{x}) \geq 0$ for all $\mathbf{x} \in \mathbf{X}$, and
  c. For any $\mathbf{x}^{-i} \in \mathbf{X}^{-i}$, a strategy $x^i$ is a best response to $\mathbf{x}^{-i}$ if and only if $F^i(x^i, \mathbf{x}^{-i}) = 0$.

The expected reward function $R^i(\mathbf{x}) = \mathbb{E}_{\mathbf{a} \sim \mathbf{x}}\left[r^i(\mathbf{a})\right]$ can be expressed as a sum of products:

$$R^i(\mathbf{x}) = \sum_{\tilde{\mathbf{a}} \in \mathbf{A}} r^i(\tilde{\mathbf{a}})\mathbb{P}_{\mathbf{a} \sim \mathbf{x}}\left(\mathbf{a} = \tilde{\mathbf{a}}\right) = \sum_{\tilde{\mathbf{a}} \in \mathbf{A}} r^i(\tilde{a}^1, \dots, \tilde{a}^n) \prod_{j=1}^{n} x^j(\tilde{a}^j), \quad \forall \mathbf{x} \in \mathbf{X}.$$

From this form, it is immediate that $R^i$ is continuous on $\mathbf{X}$. Moreover, it can easily be shown that $R^i$ is multi-linear in $\mathbf{x}$. That is, for any $j \in [n]$, fixing $\mathbf{x}^{-j}$, we have that $x^j \mapsto R^i(x^j, \mathbf{x}^{-j})$ is linear.[3]

Since $R^i$ is continuous on $\mathbf{X}$ and $\mathbb{A}^i$ is a finite set, one has that the pointwise maximum of finitely many continuous functions is continuous. Thus, the function

$$\mathbf{x}^{-i} \mapsto \max_{a^i \in \mathbb{A}^i} R^i\left(\delta_{a^i}, \mathbf{x}^{-i}\right)$$

is continuous on $\mathbf{X}^{-i}$. Since $F^i(x^i, \mathbf{x}^{-i}) = \max_{a^i \in \mathbb{A}^i} R^i\left(\delta_{a^i}, \mathbf{x}^{-i}\right) - R^i(x^i, \mathbf{x}^{-i})$ is the difference of continuous functions, $F^i$ is also continuous. This proves item a.

From the multi-linearity of $R^i$, we have that, for fixed $\mathbf{x}^{-i} \in \mathbf{X}^{-i}$, the optimization problem $\sup_{x^i \in \mathcal{X}^i} R^i(x^i, \mathbf{x}^{-i})$ is equivalent to a linear program

$$\sup_{x^i \in \mathbb{R}^{\mathbb{A}^i}} w_{\mathbf{x}^{-i}}^\top x^i, \quad \text{subject to} \begin{cases} \mathbf{1}^\top x^i = 1, \\ x^i \geq 0 \end{cases},$$

where $w_{\mathbf{x}^{-i}} \in \mathbb{R}^{\mathbb{A}^i}$ is a vector defined by $w_{\mathbf{x}^{-i}}(a^i) := R^i(\delta_{a^i}, \mathbf{x}^{-i})$.

The vertices of the feasible set for the latter linear program are precisely the points $\{\delta_{a^i} : a^i \in \mathbb{A}^i\}$. This implies that $\max_{a^i} R^i(\delta_{a^i}, \mathbf{x}^{-i}) \geq R^i(x^i, \mathbf{x}^{-i})$ for any $x^i, \mathbf{x}^{-i}$. Items b and c follow. From this formulation, one can also see that a player $i \in [n]$ is satisfied at $\mathbf{x} \in \mathbf{X}$ if and only if its strategy $x^i$ is supported on the set of maximizers $\arg\max_{a^i \in \mathbb{A}^i}\{R^i(\delta_{a^i}, \mathbf{x}^{-i})\}$.

# B    Proof of Lemma 1

Recall that in the proof of Theorem 1, $\mathbf{x}_\star$ was defined to be some strategy accessible from $\mathbf{x}_k \in \mathbf{X}$ such that all players unsatisfied at $\mathbf{x}_k$ were satisfied at $\mathbf{x}_\star$. The statement of Lemma 1 was the following.

> **Lemma 1** *If* $\mathrm{Worse}(\mathbf{x}_k) = \varnothing$, *then there exists a sequence* $\{\mathbf{y}_t\}_{t=1}^\infty$, *with* $\mathbf{y}_t \in \mathrm{NoBetter}(\mathbf{x}_k)$ *for each $t$, such that* $\lim_{t \to \infty} \mathbf{y}_t = \mathbf{x}_\star$.

*Proof.* Suppose, to the contrary, that no such sequence exists. Then, there exists some $\zeta > 0$ such that for every $\mathbf{z} \in \mathrm{Access}(\mathbf{x}_k) \cap N_\zeta(\mathbf{x}_\star)$, one has $\mathbf{z} \notin \mathrm{NoBetter}(\mathbf{x}_k)$. That is, some player unsatisfied at

---

[3]Of course, scaling inputs of $R^i$ means the resulting argument is no longer a probability vector. However, one can simply linearly extend $R^i$ to be a function on $\mathbb{R}^d$, where $d = \sum_{j=1}^n |\mathbb{A}^j|$.

$\mathbf{x}_k$ is satisfied at $\mathbf{z}$. Equivalently, for some $i \in \mathrm{UnSat}(\mathbf{x}_k)$, we have $z^i \in \mathrm{BR}_0^i(\mathbf{z}^{-i})$. This implies that for that player $i$, that value of $\zeta$, and the strategy profile $(z^i, \mathbf{z}^{-i}) \in N_\zeta(\mathbf{x}_\star)$, $z^i$ is supported on the set $\mathrm{argmax}_{a^i \in \mathbb{A}^i}\{R^i(\delta_{a^i}, \mathbf{z}^{-i})\}$.

For each $\xi \geq 0$, we define a strategy profile $\mathbf{w}_\xi \in \mathbf{X}$ as follows:

$$w_\xi^i := \begin{cases} (1 - \xi)x_k^i + \xi \mathrm{Uniform}(\mathbb{A}^i), & \text{if } i \in \mathrm{UnSat}(\mathbf{x}_k) \\ x_k^i, & \text{else.} \end{cases}$$

Note that we have defined $w_\xi^i = x_k^i$ for $i \in \mathrm{Sat}(\mathbf{x}_k)$, which is to say that we change only the strategies of the unsatisfied players, meaning $\mathbf{w}_\xi \in \mathrm{Access}(\mathbf{x}_k)$. We will show that if $\xi > 0$ is sufficiently small, then continuity of the functions $\{F^i\}_{i \in [n]}$ guarantees that $\mathbf{w}_\xi \in \mathrm{NoBetter}(\mathbf{x}_k)$.

Indeed, player $i \in [n]$ is unsatisfied at $\mathbf{x}_k$ if and only if it fails to best respond, $x_k^i \notin \mathrm{BR}_0^i(\mathbf{x}_k^{-i})$. Using the function $F^i$, this is equivalent to $F^i(x_k^i, \mathbf{x}_k^{-i}) > 0$. For each player $i \in \mathrm{UnSat}(\mathbf{x}_k)$, let $\sigma_i > 0$ be such that $F^i(\mathbf{x}_k) \geq \sigma_i > 0$. Define $\bar{\sigma} = \min\{\sigma_i : i \in \mathrm{UnSat}(\mathbf{x}_k)\}$.

The following statement holds by the continuity of the functions $\{F^i\}_{i=1}^n$: for each player $i \in [n]$, there exists $e_i > 0$ such that if a strategy profile $\mathbf{x}$ belongs to the $e_i$ neighbourhood of $\mathbf{x}_k$ (i.e. $\mathbf{x} \in N_{e_i}(\mathbf{x})$), then $|F^i(\mathbf{x}) - F^i(\mathbf{x}_k)| < \bar{\sigma}/2$. Since $F^i(\mathbf{x}_k) \geq \bar{\sigma}$, it follows that $F^i(\mathbf{x}) > \bar{\sigma}/2 > 0$, and player $i$ is not best responding at $\mathbf{x} \in N_{e_i}(\mathbf{x})$.

Let $\bar{e} := \min\{e_i : i \in [n]\}$. By taking $\xi < \bar{e}/(2n)$, one has that $\mathbf{w}_\xi \in N_{\bar{e}}(\mathbf{x}_k)$. From the preceding remarks, one can see that $\mathrm{UnSat}(\mathbf{x}_k) \subseteq \mathrm{UnSat}(\mathbf{w}_\xi)$, since all players who were unsatisfied at $\mathbf{x}_k$ remain unsatisfied at $\mathbf{w}_\xi$. Since $w_\xi^j = x_k^j$ for any player $j \in \mathrm{Sat}(\mathbf{x}_k)$, one also has that $\mathbf{w}_\xi \in \mathrm{Access}(\mathbf{x}_k)$. These two parts combine to show that $\mathbf{w}_\xi \in \mathrm{NoBetter}(\mathbf{x}_k)$.

Fixing $\xi > 0$ at a sufficiently small value ($\xi \in (0, \bar{e}/2n)$), the preceding deductions show that $\mathbf{w}_\xi \in \mathrm{NoBetter}(\mathbf{x}_k)$. By the earlier discussion, we have that $\mathbf{w}_\xi \notin N_\zeta(\mathbf{x}_\star)$.

A very important aspect of this construction is that $w_\xi^i(a^i) > 0$ for each $i \in \mathrm{UnSat}(\mathbf{x}_k)$ and action $a^i \in \mathbb{A}^i$, so that $w_\xi^i$ is fully mixed for each player who was unsatisfied at $\mathbf{x}_k$.

Next, for each $\lambda \in [0, 1]$ and player $i \in \mathrm{UnSat}(\mathbf{x}_k)$, we define

$$z_\lambda^i = (1 - \lambda)x_\star^i + \lambda w_\xi^i.$$

We also define $z_\lambda^i = x_k^i$ for players $i \in \mathrm{Sat}(\mathbf{x}_k)$. For sufficiently small values of $\lambda$, say $\lambda \leq \bar{\lambda}$, we have that $\mathbf{z}_\lambda \in N_\zeta(\mathbf{x}_\star)$, which implies $\mathbf{z}_\lambda \notin \mathrm{NoBetter}(\mathbf{x}_k)$.

This implies that there exists a player $i^\dagger \in \mathrm{UnSat}(\mathbf{x}_k)$ for whom

$$z_\lambda^{i^\dagger} \in \mathrm{BR}_0^{i^\dagger}\left(\mathbf{z}_\lambda^{-i^\dagger}\right), \text{ for infinitely many } \lambda \in \left(0, \bar{\lambda}\right].$$

(The existence of such a player is perhaps not obvious. As we previously noted, for $\lambda < \bar{\lambda}$, we have $\mathbf{z}_\lambda \notin \mathrm{NoBetter}(\mathbf{x}_k)$, which means there exists *some* player $i^\dagger(\lambda)$ that was unsatisfied at $\mathbf{x}_k$ and is satisfied at $\mathbf{z}_\lambda$. The identity of this player may change with $\lambda$. To see that some particular individual must satisfy this best response condition infinitely often, one can apply the pigeonhole principle to the set $\{\bar{\lambda}, \bar{\lambda}/2, \ldots, \bar{\lambda}/m\}$ for arbitrarily large $m$.)

By our definition of $z_\lambda^{i^\dagger}$ as a convex combination involving $\mathrm{Uniform}(\mathbb{A}^{i^\dagger})$, we have that $z_\lambda^{i^\dagger}$ is fully mixed and puts positive probability on each action in $\mathbb{A}^{i^\dagger}$. Using the characterization involving $F^{i^\dagger}$, the fact that $z_\lambda^{i^\dagger} \in \mathrm{BR}_0^{i^\dagger}\left(\mathbf{z}_\lambda^{-i^\dagger}\right)$ and the fact that $z_\lambda^{i^\dagger}$ is fully mixed together imply that $R^{i^\dagger}(\delta_a, \mathbf{z}_\lambda^{-i^\dagger}) = R^{i^\dagger}(\delta_{a'}, \mathbf{z}_\lambda^{-i^\dagger})$, for any $a, a' \in \mathbb{A}^{i^\dagger}$. This can be equivalently re-written as

$$\sum_{\mathbf{a}^{-i^\dagger}} r^{i^\dagger}(a, \mathbf{a}^{-i^\dagger}) \prod_{j \neq i^\dagger} \left\{(1 - \lambda)x_\star^j(a^j) + \lambda w_\xi^j(a^j)\right\}$$

$$= \sum_{\mathbf{a}^{-i^\dagger}} r^{i^\dagger}(a', \mathbf{a}^{-i^\dagger}) \prod_{j \neq i^\dagger} \left\{(1 - \lambda)x_\star^j(a^j) + \lambda w_\xi^j(a^j)\right\}$$

$$\Longleftrightarrow \sum_{\mathbf{a}^{-i^\dagger}} \left[r^{i^\dagger}(a, \mathbf{a}^{-i^\dagger}) - r^{i^\dagger}(a', \mathbf{a}^{-i^\dagger})\right] \prod_{j \neq i^\dagger} \left\{(1 - \lambda)x_\star^j(a^j) + \lambda w_\xi^j(a^j)\right\} = 0 \qquad (3)$$

for any $a, a' \in \mathbb{A}^{i^\dagger}$.

The lefthand side of the final equality (3) is a polynomial in $\lambda$ of finite degree, but admits infinitely many solutions (from our choice of $i^\dagger$). This implies that it is the zero polynomial. In turn, this implies that the left side of (3) holds for any $\lambda \in [0, 1]$, and in particular for $\lambda = 1$. This means $z_1^{i^\dagger} \in \mathrm{BR}_0^{i^\dagger}(\mathbf{z}_1^{-i^\dagger})$, meaning $\mathbf{z}_1 \notin \mathrm{NoBetter}(\mathbf{x}_k)$. On the other hand, we have $\mathbf{z}_1 = \mathbf{w}_\xi \in \mathrm{NoBetter}(\mathbf{x}_k)$, a contradiction.

Thus, we see that there exists a sequence $\{\mathbf{y}_t\}_{t=1}^\infty$, with $\mathbf{y}_t \in \mathrm{NoBetter}(\mathbf{x}_k)$ for all $t$, such that $\lim_{t\to\infty} \mathbf{y}_t = \mathbf{x}_\star$. $\qquad\square$

## C  Markov games: model and connections to Theorem 1

Markov games are popular model in the field of multi-agent reinforcement learning. Since the model is quite standard, we offer a short description of the fundamental objects and notations, and we then describe connections between Theorem 1 and a possible extension to multi-state Markov games.

A Markov game with $n$ players and discounted rewards is described by a list $\mathbf{G} = (n, \mathcal{S}, \mathbf{A}, \mathcal{T}, \mathbf{r}, \gamma)$, where $\mathcal{S}$ is a finite set of statess, $\mathbf{A} = \mathbb{A}^1 \times \cdots \times \mathbb{A}^n$ is a finite set of action profiles, and $\mathbf{r} = (r^i)_{i=1}^n$ is a collection of reward functions, where $r^i : \mathcal{S} \times \mathbf{A} \to \mathbb{R}$ describes the reward to player $i$. A transition probability function $\mathcal{T} \in \mathcal{P}(\mathcal{S}|\mathcal{S} \times \mathbf{A})$ governs the evolution of the state process, described below, and a discount factor $\gamma \in (0, 1)$ is used to aggregate rewards across time.

**Description of play.** Markov games are played across discrete time, indexed by $t \in \mathbb{N}$. At time $t$, the state variable is denoted $s_t \in \mathcal{S}$ and each player $i \in [n]$ selects an action $a_t^i \in \mathbb{A}^i$ according to a distribution $\pi^i(\cdot|s_t)$: $a_t^i \sim \pi^i(\cdot|s_t)$. The transition probability function $\pi^i \in \mathcal{P}(\mathbb{A}^i|\mathcal{S})$ is called player $i$'s *policy*, and we denote player $i$'s set of policies by $\Pi^i := \mathcal{P}(\mathbb{A}^i|\mathcal{S})$. For any time $t \in \mathbb{N}$, the collection of actions $\{a_t^i\}_{i=1}^n$ is mutually conditionally independent given $s_t$. Upon selection of the action profile $\mathbf{a}_t := (a_t^i)_{i=1}^n$, each player $i$ receives the reward $r^i(s_t, \mathbf{a}_t)$, and the state transitions from $s_t$ to $s_{t+1}$ according to $s_{t+1} \sim \mathcal{T}(\cdot|s_t, \mathbf{a}_t)$.

Player $i$'s performance criterion is its expected $\gamma$-discounted return, which depends on the state variable and the collective policy profile $\boldsymbol{\pi} := (\pi^1, \ldots, \pi^n)$, which we also denote by $(\pi^i, \boldsymbol{\pi}^{-i})$ to isolate player $i$'s policy. We let $\boldsymbol{\Pi} := \Pi^1 \times \cdots \times \Pi^n$ denote the set of policy profiles. For each pair $(\boldsymbol{\pi}, s) \in \boldsymbol{\Pi} \times \mathcal{S}$, player $i$'s expected $\gamma$-discounted return is given by

$$V^i(\pi^i, \boldsymbol{\pi}^{-i}, s) := \mathbb{E}_{\boldsymbol{\pi}}\left[\sum_{t=1}^\infty \gamma^{t-1} r^i(s_t, \mathbf{a}_t)\,\middle|\, s_1 = s\right],$$

where $\mathbb{E}_{\boldsymbol{\pi}}$ denotes that for every $t \geq 1$, we have that $a_t^j \sim \pi^j(\cdot|s_t)$ for each player $j \in [n]$ and, implicitly, $s_{t+1} \sim \mathcal{T}(\cdot|s_t, \mathbf{a}_t)$.

**Definition 6.** *For $\epsilon \geq 0$, a policy $\pi_*^i \in \Pi^i$ is called an $\epsilon$-best response to $\boldsymbol{\pi}^{-i}$ if*

$$V^i(\pi_*^i, \boldsymbol{\pi}^{-i}, s) \geq V^i(\pi^i, \boldsymbol{\pi}^{-i}, s) - \epsilon, \quad \forall \pi^i \in \Pi^i, \quad \forall s \in \mathcal{S}.$$

**Definition 7.** *For $\epsilon \geq 0$, a policy profile $\boldsymbol{\pi}_* = (\pi_*^i, \boldsymbol{\pi}_*^{-i}) \in \boldsymbol{\Pi}^i$ is called a* Markov perfect $\epsilon$-equilibrium *if, for each player $i \in [n]$, $\pi_*^i$ is an $\epsilon$-best response to $\boldsymbol{\pi}_*^{-i}$.*

Putting $\epsilon = 0$ into the definitions above, we recover the classical definitions of best responding and Markov perfect equilibrium. In analogy to normal-form games, we use $\mathrm{BR}_\epsilon^i(\boldsymbol{\pi}^{-i}) \subseteq \Pi^i$ to denote player $i$'s set of $\epsilon$-best-responses to a given counterplayer policy profile $\boldsymbol{\pi}^{-i}$.

### Remarks on Markov games

As is conventional in the literature on MARL, we focus on policies that are stationary, Markovian, and possibly randomized. That is, we focus on policies for player $i$ that map states $s_t \in \mathcal{S}$ to distributions over the agent's action set $\mathbb{A}^i$ and sample each action $a_t^i$ from that distribution in a time-invariant and history-independent manner. In principle, agents could use policies that depend also on the time index $t$ or on the history of states and actions. However, the bulk of works on MARL consider this simpler class of policies and this is justifiable for several reasons. We refer the reader to [30] for a summary of such justifications.

Markov games generalize both normal-form games (taking the state space $\mathcal{S}$ to be a singleton) and also MDPs (taking the number of players $n = 1$). Moreover, when player $i$'s counterplayers follow a stationary policy $\boldsymbol{\pi}^{-i} \in \boldsymbol{\Pi}^{-i}$, as assumed in this work, player $i$'s stochastic control problem is equivalent to a single-agent MDP (whose problem data depend on $\boldsymbol{\pi}^{-i}$). It follows that player $i$'s set of (stationary) best responses to $\boldsymbol{\pi}^{-i}$ is non-empty. Furthermore, player $i$'s best response condition can be characterized using the familiar action value (Q-) functions of reinforcement learning theory. We briefly summarize this below.

In addition to the objective criterion $V^i(\pi^i, \boldsymbol{\pi}^{-i}, s)$, which is called the value function, we may also define the action value function $Q^i$ for player $i$ as

$$Q^i(\pi^i, \boldsymbol{\pi}^{-i}, s, a^i) := \mathbb{E}_{\boldsymbol{\pi}} \left[ \sum_{t=1}^{\infty} \gamma^{t-1} r^i(s_t, \mathbf{a}_t) \middle| s_1 = s, a_1^i = a^i \right],$$

for $(\pi^i, \boldsymbol{\pi}^{-i}) \in \boldsymbol{\Pi}, (s, a^i) \in \mathcal{S} \times \mathbb{A}^i$.

We further define an optimal action value function for player $i$ against $\boldsymbol{\pi}^{-i}$, denoted $Q_{\boldsymbol{\pi}^{-i}}^{*i}$, as

$$Q_{\boldsymbol{\pi}^{-i}}^{*i}(s, a^i) := \max_{\pi_*^i \in \Pi^i} Q^i(\pi_*^i, \boldsymbol{\pi}^{-i}, s, a^i), \quad \forall (s, a^i) \in \mathcal{S} \times \mathbb{A}^i.$$

For any policy $\boldsymbol{\pi} = (\pi^i, \boldsymbol{\pi}^{-i})$, one can express player $i$'s value function using its Q-function and conditional expectations as $V^i(\boldsymbol{\pi}, s) = \sum_{a^i} \pi^i(a^i|s) Q^i(\boldsymbol{\pi}, s, a^i)$. From this, it follows that

$$\max_{a^i \in \mathbb{A}^i} Q_{\boldsymbol{\pi}^{-i}}^{*i}(s, a^i) = \max_{\pi_*^i \in \Pi^i} V^i(\pi_*^i, \boldsymbol{\pi}^{-i}, s), \quad \forall s \in \mathcal{S}.$$

This equality allows us to characterize best responses using a function $f^i : \boldsymbol{\Pi} \to \mathbb{R}$, analogous to the function $F^i$ appearing in the normal-form case. We define $f^i(\boldsymbol{\pi})$ as

$$f^i(\pi^i, \boldsymbol{\pi}^{-i}) = \max_{s \in \mathcal{S}} \left[ \max_{a_\star^i \in \mathbb{A}^i} Q_{\boldsymbol{\pi}^{-i}}^{*i}(s, a_\star^i) - V^i(\boldsymbol{\pi}, s) \right], \quad \forall \boldsymbol{\pi} \in \boldsymbol{\Pi}.$$

The functions $\{f^i\}_{i=1}^n$ defined above possess the three properties we required of the functions $\{F^i\}_{i=1}^n$ in the proof of Theorem 1: (a) $f^i$ is continuous on $\boldsymbol{\Pi}$ [52], (b) $f^i(\boldsymbol{\pi}) \geq 0$ for all $\boldsymbol{\pi} \in \boldsymbol{\pi}$, and (c) $f^i(\pi^i, \boldsymbol{\pi}^{-i}) = 0$ if and only if $\pi^i$ is a best response to $\boldsymbol{\pi}^{-i}$.

**On extending Theorem 1 to Markov games**

We now turn our attention to the task of extending Theorem 1 to Markov games. Following the proof of Theorem 1, virtually all steps can be reproduced in the multi-state setting. To begin, one can construct a satisficing path $\boldsymbol{\pi}_1, \boldsymbol{\pi}_2, \ldots, \boldsymbol{\pi}_k$ by growing the set of unsatisfied players at each iteration until either $\mathrm{UnSat}(\boldsymbol{\pi}_k) = [n]$ or $\mathrm{Worse}(\boldsymbol{\pi}_k) = \varnothing$. In the latter case, one can consider the subgame involving only the players in $\mathrm{UnSat}(\boldsymbol{\pi}_k)$ and obtain a Markov perfect equilibrium $\tilde{\boldsymbol{\pi}}_\star$ for that subgame, which can then be extended to a policy profile $\boldsymbol{\pi}_\star \in \mathrm{Access}(\boldsymbol{\pi}_k)$ by putting

$$\pi_\star^i = \begin{cases} \tilde{\pi}_\star^i, & \text{if } i \in \mathrm{UnSat}(\boldsymbol{\pi}_k), \\ \pi_k^i, & \text{if } i \in \mathrm{Sat}(\boldsymbol{\pi}_k). \end{cases}$$

To complete the extension of Theorem 1 to Markov games, one must show that this policy $\boldsymbol{\pi}_\star \in \boldsymbol{\Pi}$ is a Markov perfect equilibrium of the $n$-player Markov game. Since the functions $\{f^i\}_{i=1}^n$ also satisfy the continuity and semi-definiteness properties described in Appendix A, one possible technique for completing this proof involves showing that the policy $\boldsymbol{\pi}_\star$ is a limit point of the set $\mathrm{NoBetter}(\boldsymbol{\pi}_k)$. In other words, one possible technique for completing this proof requires extending Lemma 1 to the multi-state case.

Up to this point, analysis of the stateless case and the multi-state case have been conducted perfectly in parallel. However, it is in the extension of Lemma 1 that the presence of a state leads to a discrepancy in the analysis that will necessitate a novel proof technique for the extension of Theorem 1 to Markov games. We elaborate below on this discrepancy.

**Normal-form game analysis.** In the context of finite normal-form games, our proof of Lemma 1 in Appendix B involves a proof by contradiction that exploits the explicit form of an indifference condition in the stateless case. In simple terms, if a player $i$ is best responding *and* placing positive probability on every action, then any two actions offer equal expected payoff. In symbols, we note that $R^i(\delta_{a_1^i}, \mathbf{x}^{-i}) = R^i(\delta_{a_2^i}, \mathbf{x}^{-i})$ if and only if

$$\sum_{\mathbf{a}^{-i} \in \mathbf{A}^{-i}} \left[ r^i(a_1^i, \mathbf{a}^{-i}) - r^i(a_2^i, \mathbf{a}^{-i}) \right] \mathbb{P}_{\mathbf{x}^{-i}}(\mathbf{a}^{-i})$$
$$= \sum_{\mathbf{a}^{-i} \in \mathbf{A}^{-i}} \left[ r^i(a_1^i, \mathbf{a}^{-i}) - r^i(a_2^i, \mathbf{a}^{-i}) \right] \prod_{j \neq i} \left\{ x^j(a^j) \right\} = 0.$$

For reasons that will be clarified below, we refer to the expressions $\left[ r^i(a_1^i, \mathbf{a}^{-i}) - r^i(a_2^i, \mathbf{a}^{-i}) \right]$ as *coefficient terms*, and we refer to the terms $\mathbb{P}_{\mathbf{x}^{-i}}(\mathbf{a}^{-i}) = \prod_{j \neq i} \left\{ x^j(a^j) \right\}$ as strategy-dependent terms. We remark that in the case of normal-form games, the coefficient terms above do not depend on the strategy $\mathbf{x}^{-i}$.

Our proof of Lemma 1 in Appendix B considered a one-parameter family of strategies parameterized by $\lambda \in [0, 1]$. As part of an intricate proof by contradiction, we obtained an indifference condition, (3), for a player $i^\dagger$ who played each action with positive probability while also best responding. Due to the explicit parameterization by $\lambda$ of the strategy $\mathbf{z}_\lambda$, we are able to recognize that the indifference condition in (3) is characterized by the roots of a polynomial in $\lambda$. Critically, the lefthand-side of (3) is a polynomial in $\lambda$ because the coefficient terms do not depend on the strategy $\mathbf{z}_\lambda$ and hence do not depend on $\lambda$, while the strategy-dependent terms are polynomials in $\lambda$.

**Markov game analysis.** By contrast, we now study indifference conditions in Markov games. Consider an agent $i$ who is best responding to a policy $\boldsymbol{\pi}^{-i}$ and places positive probability on actions $a_1^i$ and $a_2^i$ in state $s$. The optimality condition is turned into an indifference condition between $a_1^i$ and $a_2^i$ in state $s$ as follows:

$$Q_{\boldsymbol{\pi}^{-i}}^{*i}(s, a_1^i) = Q_{\boldsymbol{\pi}^{-i}}^{*i}(s, a_2^i) = \max_{a^i \in \mathbb{A}^i} Q_{\boldsymbol{\pi}^{-i}}^{*i}(s, a^i).$$

One can show that $Q_{\boldsymbol{\pi}^{-i}}^{*i}$ satisfies the following equality for any $(s, a^i) \in \mathcal{S} \times \mathbb{A}^i$:

$$Q_{\boldsymbol{\pi}^{-i}}^{*i}(s, a^i) = \sum_{\mathbf{a}^{-i} \in \mathbf{A}^{-i}} \left[ r^i(s, a^i, \mathbf{a}^{-i}) + \gamma \sum_{s' \in \mathcal{S}} \mathcal{T}(s'|s, a^i, \mathbf{a}^{-i}) \max_{a_\star^i \in \mathbb{A}^i} Q_{\boldsymbol{\pi}^{-i}}^{*i}(s', a_\star^i) \right] \mathbb{P}_{\boldsymbol{\pi}}(\mathbf{a}^{-i}|s),$$

where $\mathbb{P}_{\boldsymbol{\pi}}(\mathbf{a}^{-i}|s) = \prod_{j \neq i} \pi^j(a^j|s)$ denotes the probability of the action profile $\mathbf{a}^{-i}$ in state $s$ under policy $\boldsymbol{\pi}$. In analogy to the normal-form case, we refer to $\mathbb{P}_{\boldsymbol{\pi}}(\mathbf{a}^{-i}|s)$ as the strategy-dependent term and we refer to the term enclosed in square brackets as the coefficient term. However, unlike the normal-form case, here it is clear that the (so-called) coefficient term also depends on the policy $\boldsymbol{\pi}^{-i}$, through the term $\max_{a_\star^i \in \mathbb{A}^i} Q_{\boldsymbol{\pi}^{-i}}^{*i}(s, a_\star^i)$.

Suppose now that we obtain a one-parameter family of policies $\{\boldsymbol{\varpi}_\lambda : 0 \leq \lambda \leq 1\}$ parameterized by some $\lambda \in [0, 1]$, in analogy to our construction of $\mathbf{z}_\lambda$ in Appendix B. Since the coefficient term depends on the policy of player $i$'s counterplayers, one has that the indifference condition

$$Q_{\boldsymbol{\varpi}_\lambda^{-i}}^{*i}(s, a_1^i) - Q_{\boldsymbol{\varpi}_\lambda^{-i}}^{*i}(s, a_2^i) = 0$$

cannot generally be characterized by the roots of a polynomial in the parameter $\lambda$.[4]

Without characterization of the indifference condition as a polynomial in the policy parameter, our proof technique in Appendix B becomes unsuitable for the multi-state setting: we cannot invoke the fundamental theorem of algebra to conclude that the coefficient terms are identically zero, and thus we cannot obtain the contradiction critical to our proof by contradiction, where we found that player $i^\dagger$ is in fact indifferent even at the extreme parameter value of $\lambda = 1$.

---

[4]Although this indifference condition does not generally yield a polynomial in $\lambda$, one can easily find special cases of Markov games in which it does. For instance, if player $i$'s action does not influence transition probabilities, the indifference condition will yield a polynomial and the normal-form proof technique will go through without modification.

In summary, the proof technique employed in Appendix B to prove Lemma 1 relies crucially on the specific explicit form of the indifference condition in stateless, finite normal-form games. Passing to the multi-state setting, the analogous indifference condition takes a different form, and so the specific derivations cannot be repurposed for a simple extension of Lemma 1. However, it is also important to recognize that this phenomenon is a limitation of the proof technique and does not pose a fundamental obstacle to the generalization of Theorem 1 *per se*. Indeed, the remaining elements of the proof of Theorem 1 carry over seemlessly to the multi-state case, including various continuity conditions for functions characterizing best responses. It therefore seems promising that one can generalize Theorem 1 to apply to Markov games by applying similar machinery as used in this paper but substituting a different proof for that of Lemma 1 to take advantage of topological or geometric structure shared by both normal-form and Markov games. We leave this as an interesting open question for future research.

