# OpenReview forum: "Paths to Equilibrium in Games"
_NeurIPS.cc/2024/Conference — NeurIPS 2024 spotlight_

### Official Review · Reviewer_HTV5 · 2024-06-30

**Soundness:** 3
**Presentation:** 4
**Contribution:** 4
**Rating:** 6
**Confidence:** 4

**Summary:**

The authors study and affirm the possibility of constructing a satisficing path to a Nash equilibrium, from any initial strategy profile, in n-player normal-form games. Satisficing paths generalize best-response paths in that a player that is not best responding is not restricted in its strategy update. Players that were previously best-responding maintain their strategy. The authors use their results in a discussion to suggest that future design of MARL algorithms incorporate exploration periods among strategies, especially when they are not best-responding. The authors go on to discuss their contributions in the contexts of Markov games, decentralized learning, and dynamical systems.

**Strengths:**

The paper is well-described and well thought-out. Satisficing improvements is an interesting concept and could be useful in MARL research. The main contribution (Theorem 1) seems to provide a useful conclusion about how MARL algorithms could interface with satisficing paths rather than the best-response concept. Good extension beyond the work of (51) (line 183). It's good that the proof is not immediate, especially with Lemma 1 in Appendix A. Good proof sketch on Line 189. Overall, I believe this paper would provide a fruitful conceptual addition to the MARL research discussion.

**Weaknesses:**

Remark 1 suggests the algorithm is centralized rather than decentralized and learning-based. Under Case 1 you say "fix an arbitrary Nash equilibrium $z^*$.'' How does one discover "z^*?" Isn't the point of using MARL algorithms to (1) use decentralized learning to discover a NE, or (2) identify that NE from a centralized perspective, which is PPAD-complete? I am confused how the centralized entity forms its recommendations for which strategies agents should play next.

A similar confusion holds for Case 2: I could be reading this not clearly enough, but the proof appears to suggest ``make most agents unsatisfied then arbitrarily jump to a NE (subject to feasibility via the remaining satisficed agents).'' Could you please elaborate how this is not a trivial conclusion (line 367)?

**Questions:**

If there's enough space, could you elaborate on the open question left by (51) on line 73? It seems that the satisficing principle is significantly studied (line 55) but I don't have the best understanding for how or why this principle is a significant in the pre-existing literature.

**Limitations:**

Yes

---

> ### Author Rebuttal · Authors · 2024-08-06
>
> Thank you for your close reading of our paper and for sharing your concerns. We wish to primarily address the points of confusion identified in your review, but perhaps it would be helpful to first address your question on the connection between our work and the previous literature.
>
> Indeed, the satisficing principle has been widely employed in several previous works across game theory and MARL, such as those identified on Line 55. While the principle itself is natural and widely used, the graph theoretic structure was studied only indirectly by early proponents of satisficing algorithms (e.g. [19], [20]), and the terminology of satisficing was not introduced until later. Since this idea appeared inherently in earlier algorithms, it is somewhat well studied, with some interesting theorems proved incidentally while studying convergence properties of specific algorithms. On the other hand, since it was not formalized until later, some fundamental structural questions (such as the open question we answer) were not addressed head-on in those earlier works.
>
> Complementing and building on existing work, our theoretical study on the structure of satisficing paths may be interpreted as being upstream of the analysis of specific MARL algorithms. One aim of our work was to show that such paths to equilibrium exist, since this existence result is necessary for any satisficing-based MARL algorithm to be effective. That being said, it is important to note that the structure we study is separate from any individual satisficing-based MARL algorithm. In fact, the process described in the Proof of Theorem 1 is not an algorithm, but an existence proof described procedurally. Remark 1 cautions the reader that each step of this procedure is analytic rather than algorithmic. As an existence proof, justified non-constructive steps (e.g. the selection of an arbitrary Nash equilibrium z^{*} by the analyst, which is guaranteed to exist) are acceptable, and questions of complexity of such non-constructive steps are not applicable as it relates to an existence proof such as ours. Nevertheless, these questions are certainly relevant and important for downstream applications such as the complexity analysis of specific MARL algorithms.

---

> > ### Comment · Reviewer_HTV5 · 2024-08-10
> > **Thank you for your response.**
> >
> > Thank you for your response.

---

### Official Review · Reviewer_1c5w · 2024-07-01

**Soundness:** 4
**Presentation:** 3
**Contribution:** 3
**Rating:** 6
**Confidence:** 2

**Summary:**

This paper explores the dynamics of strategy updates in MARL and game theory, focusing on sequences of strategies satisfying a pairwise constraint. This constraint requires that the agent best responding in one period does not switch its strategy in the next period but does not constrain the non-optimizing agents in any way. Sequences of strategy with this property are called satisficing paths. In this paper, authors have shown that every finite normal-form game enjoys the satisficing paths property. Overall, the paper provides a new perspective on achieving equilibrium in multi-agent systems and has significant implications for the design and analysis of MARL algorithms.

**Strengths:**

This paper studies a very interesting problem: is it always possible to construct a satisficing path that terminates at equilibrium for a given game and initial strategy profile? The main strengths of this paper are:
1. The introduction of satisficing paths as a generalization of the best response path is both novel and insightful.

2. The authors provide rigorous proof that any finite normal-form game has the satisficing paths property. This proof is well-structured and addresses an open question in the field.

3. The paper’s finding that reward-deteriorating strategic updates can drive play to equilibrium is counterintuitive yet compelling.

4. The practical implications for the design of MARL algorithms are articulated. The paper provides concrete suggestions on how to incorporate satisficing updates and exploratory strategies, which could improve convergence properties and performance in a wider range of games.

**Weaknesses:**

1. Although this paper provides extensive theoretical results, it lacks empirical validation. Including experimental results or simulations, even a simple example, to demonstrate the practical applicability and performance of satisficing paths in real-world scenarios would strengthen the paper’s impact.

2. While the theoretical contributions are significant, the proof techniques used are complex and may not be easily accessible to all readers. Providing more intuitive explanations could make the results more understandable to a broader audience. Although it is not a large weakness, it would improve the readability if it can be resolved.

3. The paper offers valuable algorithmic insights but falls short of providing detailed guidance on implementing these insights in practical MARL algorithms. Including specific case studies would make the theoretical contributions more actionable for practitioners.

**Questions:**

1.	This paper provides extensive theoretical results and these results are compelling, have authors considered validating findings through empirical experiments? I am wondering how satisficing paths perform in practical multi-agent reinforcement learning scenarios.

2.	Authors have proved that any finite normal-form games have the satisficing paths property. How do you envision the scalability of your approach to larger, more complex game settings? Are there specific scalability challenges that need to be addressed?

3.	Here, I just wondering have authors considered exploring alternative pathways or strategies for reaching equilibrium. If yes, how do these alternatives compare to satisficing paths in terms of convergence speed and robustness?

**Limitations:**

Yes, the authors provided limitations.

---

> ### Author Rebuttal · Authors · 2024-08-06
>
> Thank you for your kind words and for your questions. Efficiency and practical performance are important considerations for work on MARL. Our theoretical study here is somewhat orthogonal to efficiency, and we do not take a position on the exploratory ("lose-shift") mechanism used by any particular satisficing-based MARL algorithm, but we can say a few things about the connections between this theory and efficiency.
>
> 1. Satisficing paths can be produced as the output of many different MARL algorithms. In fact, they can be produced by any algorithm that respects the "win-stay" condition of keeping an old strategy whenever it is a best response to the last period's strategy profile. This means a wide class of algorithms are available to us for simulation purposes, and a given algorithm may be highly effective (or highly ineffective) for a given game. As such, we believe that simulations of a particular MARL algorithm in a specific game may not be a meaningful representation of the theoretical findings of this paper, on the existence of paths.
>
> 2. In [20], the authors studied a particular satisficing algorithm that relied purely on exploration to drive play Nash equilibria. The authors of that work offered some remarks on how their algorithm scales with the number of agents and the size of their action sets. In the case of *purely* randomized exploration when players are unsatisfied, the authors observed that their algorithm scales poorly and inefficiently. Their observations underline the fact that a well-selected strategy update function is needed to ensure good performance in larger, more complex games.
>
> 3. We have considered a spectrum of pathways for reaching equilibrium, with one end of the spectrum being pure best response dynamics (possibly with inertia) and the other end being purely exploratory strategy updates (when players are unsatisfied). In terms of convergence speed, dynamics based on best responding without exploration can be very quick in some settings, such as potential games, but can also fail to converge in other settings, such as zero-sum games. Thus, algorithms based on exploration with the satisficing principle tend to be more robust, if slower, than algorithms on the other end of the spectrum. Of course, there are other paradigms for algorithm design outside of this spectrum, which we believe are also very interesting.

---

> > ### Comment · Reviewer_1c5w · 2024-08-12
> >
> > Thank you for your detailed responses. I keep the score.

---

### Official Review · Reviewer_sDku · 2024-07-02

**Soundness:** 4
**Presentation:** 4
**Contribution:** 3
**Rating:** 6
**Confidence:** 4

**Summary:**

The paper answers the following question affirmatively: "For an arbitrary n-player normal-form game and an arbitrary initial strategy profile x1, is it always possible to construct a satisficing path from x1 to a Nash equilibrium of the game?".

**Strengths:**

The presentation of the paper is excellent, in particular it is very clear which research question is tackled by the authors. The proof of the main result (including lemmas) is not trivial and offers interesting elements.

**Weaknesses:**

I have one main complaint, and this is related to my disagreement with the following sentence at the beginning of section 3.2:

``Theorem 1 shows that play will be driven along satisficing paths to an equilibrium".''

I disagree, because to have the guarantee that it will be driven to an equilibrium, you still need to find a way to get all unsatisfied players to the equilibrium, and this is not a trivial task at all. What your theorem 1 shows is the existence of such path to equilibrium. However, you construct this path in a very specific way, and even in the case where all players are unsatisfied (that is your Case 1, page 6), you need to find a way to transition from this situation of all-unsatisfied-players to an equilibrium (Case 1, page 6 assumes that you know such an equilibrium). And finding this is not easy if you don't know an equilibrium a priori. In other words: in large games, it may occur that almost every path is satisficing because almost no player will be exactly satisfied. This makes me wonder about the usefulness of the concept of satisficing path. The idea of weakening best-response paths where all players best-respond is interesting, but the current definition of satisficing path may be too weak.

To summarize: the paper presents an existence result, but it would be much stronger and more meaningful if authors presented an algorithm to reach Nash equilibria along satisficing paths without knowing the Nash equilibrium a priori.

**Questions:**

cf. my comments in the "Weaknesses" section.

**Limitations:**

yes

---

> ### Author Rebuttal · Authors · 2024-08-06
>
> Thank you for your engaged reading and review of our paper!
>
> Our aim with the original wording at the beginning of Section 3.2 was to contrast approaches based on satisficing with approaches based (for instance) on best responding, where in the latter it is possible that no path to equilibrium exists from a particular starting point. In the revised version, Section 3.2 will begin with the following: "When coupled with a MARL algorithm that uses an exploratory satisficing strategy update, play will be driven along satisficing paths. Theorem 1 shows that for any starting strategy profile, some such path connects the strategy profile to an equilibrium, and so a sufficiently exploratory strategy update may drive play to equilibrium along a satisficing path."
>
> With regards to a point raised in your summary, we wish to note that algorithms based (implicitly) on the satisficing principle with sufficient exploratory search when unsatisfied were proposed in references [19] and [20]. Those algorithms were restricted to special classes of games less general than those considered here (namely, two-player games in [19] and games satisfying regularity conditions in [20]). Those algorithms discretize each player's strategy set and select new strategies from this discrete subset. The satisfaction condition used in these papers replaced best responding by approximate best responding, and the authors showed that, in their specific subclasses of games, MARL using discretized satisficing leads to approximate equilibrium with arbitrarily high probability. One of the objectives of our paper was to show that such algorithms actually have theoretical basis beyond the settings considered in [19] and [20]. We believe Theorem 1 serves this objective and offers an interesting perspective on its own.  As a topic of future work, we agree that it would be interesting to study how the existence of paths interfaces with discretization and approximate best responding, which were jointly needed for the algorithmic guarantees of [19] and [20].

---

> ### Comment · Reviewer_sDku · 2024-08-12
>
> I have read the authors' rebuttal and keep my score unchanged.

---

### Official Review · Reviewer_a3ET · 2024-07-15

**Soundness:** 3
**Presentation:** 2
**Contribution:** 3
**Rating:** 6
**Confidence:** 2

**Summary:**

The paper delves into the strategic dynamics of MARL, focusing on the evolution of strategy profiles among agents. It introduces satisficing paths, a sequence of strategies where an agent that is best responding in one period does not switch strategies in the next, which allows for exploration in optimization. The central question addressed is whether, for any given game and initial strategy profile, it's always possible to construct a satisficing path that leads to Nash Equilibrium. The paper provides a positive answer for normal-form games, suggesting that reward-deteriorating strategic updates can drive play towards Nash equilibrium. The analysis has implications for MARL algorithms, which offers new insights into algorithm design that could avoid common drawbacks like cyclical behaviour.

**Strengths:**

1. The paper provides a significant theoretical result regarding the existence of satisficing paths in normal-form games.
2. It offers a novel perspective on how suboptimal strategic updates can aid convergence to equilibrium, contrary to typical reward-improving approaches.
3. The findings have clear implications for MARL algorithm design, suggesting potential modifications to enhance convergence properties.
4. The results are not limited to two-player games but extend to n-player general-sum normal-form games.

**Weaknesses:**

1. The paper is theoretical, and it's unclear how the insights would translate into practical MARL algorithms without empirical validation.

**Questions:**

1. How can the theoretical results be practically implemented in MARL algorithms?
2. Are there any known limitations or exceptions where satisficing paths might not lead to equilibrium?
3. Can the concepts presented be extended to continuous action spaces or other game types?

**Limitations:**

See above comments.

---

> ### Author Rebuttal · Authors · 2024-08-06
>
> 1. Our theoretical results have some practical consequences for MARL algorithms. First, they can be used to justify and analyze existing algorithms (such as those of references [19], [20], [33], and others) beyond the narrower classes of games for which they were designed. Second, our results inform algorithm design in MARL to break cycles, as described in Lines 298-308 of our paper. In particular, one can combine random search when unsatisfied with cycle-breaking random search (or otherwise selective, non-best-responding strategic updates). These added ingredients will influence the trajectory of strategy iterates to follow satisficing paths, which we have shown have the potential to lead to equilibrium even in cases where the smaller set of best-response paths do not lead to equilibrium.
>
> 2. Thank you for this important question. Yes indeed, satisficing paths to equilibrium do not exist in all game theoretic settings, and negative results do exist. In this paper, we show that *mixed extensions* of  finite, normal-form games (where strategies can be probability distributions over a set of actions) always admit at least one satisficing path terminating at equilibrium from any initial strategy profile. On the other hand, if strategies are not allowed to be randomized, then one can produce examples of games with a pure Nash equilibrium and an initial strategy profile that is not connected to the Nash equilibria by a satisficing path.
>
> 3. Absolutely, the definitions of satisficing can be extended, but the proof techniques for Theorem 1 may not carry over automatically. Some details of our proofs rely on linear programming formulations where the feasible set is an n-dimensional simplex and so admits a finite number of extreme points. Our proof subsequently took advantage of this fact. When one considers continuous action spaces, the number of extreme points may not be finite, and this loss of finiteness may prevent our subsequent analysis. Although our proof technique may not immediately carry over in this more general setting, we believe similar ideas can be carried out in a generalization of this work.

---

> > ### Comment · Reviewer_a3ET · 2024-08-11
> > **Thank you for your response.**
> >
> > Thank you for your response. I keep the score.

---

### Author Rebuttal · Authors · 2024-08-06

Dear AC and reviewers,

We would like to thank you all for your time and effort in reading and reviewing our paper, and we would like to thank the reviewers for their thoughtful input and positive evaluations. For your convenience, we have responded to each reviewer's questions separately below.

Kind regards,
The Authors.

---

### Decision · Program_Chairs · 2024-09-25

**Decision:**

Accept (spotlight)

**Comment:**

All reviewers see the novelty and usefulness of the main contribution of the work, which establishes that "satisficing paths" can always be used to find equilibrium in normal-form games.